# Effect of a Lens Protein in Low-Temperature Culture of Novel Immortalized Human Lens Epithelial Cells (iHLEC-NY2)

**DOI:** 10.3390/cells9122670

**Published:** 2020-12-11

**Authors:** Naoki Yamamoto, Shun Takeda, Natsuko Hatsusaka, Noriko Hiramatsu, Noriaki Nagai, Saori Deguchi, Yosuke Nakazawa, Takumi Takata, Sachiko Kodera, Akimasa Hirata, Eri Kubo, Hiroshi Sasaki

**Affiliations:** 1Department of Ophthalmology, Kanazawa Medical University, Ishikawa 920-0293, Japan; naokiy@kanazawa-med.ac.jp (N.Y.); s-takeda@kanazawa-med.ac.jp (S.T.); n-hatsu@kanazawa-med.ac.jp (N.H.); kuboe@kanazawa-med.ac.jp (E.K.); 2Research Promotion and Support Headquarters, Fujita Health University, Aichi 470-1192, Japan; norikoh@fujita-hu.ac.jp; 3Faculty of Pharmacy, Kindai University, Osaka 577-8502, Japan; nagai_n@phar.kindai.ac.jp (N.N.); 2045110002h@kindai.ac.jp (S.D.); 4Division of Hygienic Chemistry, Faculty of Pharmacy, Keio University, Tokyo 105-8512, Japan; nakazawa-ys@pha.keio.ac.jp; 5Radiation Biochemistry, Division of Radiation Life Science, Institute for Integrated Radiation and Nuclear Science, Kyoto University, Osaka 590-0494, Japan; takumi@rri.kyoto-u.ac.jp; 6Department of Electrical and Mechanical Engineering, Nagoya Institute of Technology, Aichi 466-8555, Japan; kodera@nitech.ac.jp (S.K.); ahirata@nitech.ac.jp (A.H.); 7Center of Biomedical Physics and Information Technology, Nagoya Institute of Technology, Aichi 466-8555, Japan

**Keywords:** immortalized human lens epithelial cell clone NY2, iHLEC-NY2, low-temperature culture, computer simulation, environmental temperature, crystalline lens temperature, αA crystallin, amyloid β

## Abstract

The prevalence of nuclear cataracts was observed to be significantly higher among residents of tropical and subtropical regions compared to those of temperate and subarctic regions. We hypothesized that elevated environmental temperatures may pose a risk of nuclear cataract development. The results of our in silico simulation revealed that in temperate and tropical regions, the human lens temperature ranges from 35.0 °C to 37.5 °C depending on the environmental temperature. The medium temperature changes during the replacement regularly in the cell culture experiment were carefully monitored using a sensor connected to a thermometer and showed a decrease of 1.9 °C, 3.0 °C, 1.7 °C, and 0.1 °C, after 5 min when setting the temperature of the heat plate device at 35.0 °C, 37.5 °C, 40.0 °C, and 42.5 °C, respectively. In the newly created immortalized human lens epithelial cell line clone NY2 (iHLEC-NY2), the amounts of RNA synthesis of αA crystallin, protein expression, and amyloid β (Aβ)1-40 secreted into the medium were increased at the culture temperature of 37.5 °C compared to 35.0 °C. In short-term culture experiments, the secretion of Aβ1-40 observed in cataracts was increased at 37.5 °C compared to 35.0 °C, suggesting that the long-term exposure to a high-temperature environment may increase the risk of cataracts.

## 1. Introduction

The crystalline lens is surrounded by a lens capsule containing a large amount of type IV collagen; this capsule functions as the basement membrane of lens epithelial cells (LECs) arranged in a single layer at the anterior zone. In vivo, the tissue stem cells of LECs at the germinative zone proliferate at a low level in the bow area near the equatorial segment of the crystalline lens [1], and the cytoplasm extends toward the anterior and posterior poles of the lens after separation from the lens capsule, resulting in lens fiber cells. These cells then surround the lens fetal nucleus formed at the developmental stage while further differentiating into lens fibers.

The human crystalline lens with the fetal nucleus in the central zone maintains the transparency and flexibility for focus adjustment over the course of decades. Presbyopia occurs due to a decrease in accommodation power by sclerosis of the crystalline lens nucleus with aging; this is also thought to be a precursor of nuclear cataract. In their investigation of the relationship between environmental temperature and the age at presbyopia onset, Miranda et al. observed that individuals who live in geographic regions with higher temperatures develop presbyopia earlier [2]. Thus, if the high environmental temperature is involved in the onset of presbyopia, the high environmental temperature may be also involved in nuclear cataracts. In other words, in the tropical and subtropical regions with a high prevalence of nuclear cataracts, the ambient ultraviolet (UV) intensity is strong and the environmental temperature is high. Therefore, the environmental temperature may pose a risk of nuclear cataract development.

In an animal model, 6-month exposure to low-dose UV-A induced nuclear cataracts with opacification in the central zone of the crystalline lens [3]. However, few reports have investigated the relationship between UV-B and nuclear cataracts [4,5,6,7]. There have been at least two epidemiological studies on the prevalence of nuclear cataracts in general populations aged ≥50 in regions with different environmental conditions. These studies showed that the prevalence of nuclear cataracts of level ≥1 by the World Health Organization (WHO) cataract grading system was significantly higher in tropical and subtropical regions (i.e., Mkuranga, Tanzania; Singapore; Amami, Kagoshima, Japan; and Sanya, Hainan, China) compared to temperate regions (Monzen, Ishikawa, Japan; Taiyuan, Shanxi, China; and Shenyang, Liaoning, China) and a subarctic region (Reykjavik, Iceland), regardless of race [8,9]. The prevalence of nuclear cataracts may have increased in tropical and subtropical areas because the ambient UV intensity is stronger in low-latitude areas than that in mid- and high-latitude areas. Miyashita et al. also observed a significant association between cumulative ocular UV exposure and nuclear cataracts in the Han people [9]. In a related study, we demonstrated that high ocular UV exposure was a factor in the higher prevalence of nuclear cataracts in tropical and subtropical regions [4].

We thus hypothesized that the incidence of cataracts is related to the environmental temperature. In the present study, we evaluated the relationship between the environmental temperature and the lens temperature by conducting a computer simulation in silico. We also examined the effects of different temperatures on the lens in vitro using LECs at the lens temperature of residents of tropical and temperate regions. As LECs are extremely difficult to maintain and culture for a long period, immortalized LEC lines have been used in many experimental reports. One of the existing immortalized LEC lines, SRA01/04, was obtained from infants who underwent surgery for retinopathy of prematurity [10]. Herein, we created a vector that can introduce an immortalizing gene into the genome-specific domain by means of a new technique using primary cultured human LECs, and we used this vector to establish a new immortalized human LEC line. We cultured the new LECs at different temperatures and compared the values of amyloid-beta (Aβ, which was reported to be detected in crystalline lens protein and cataracts [11,12]), using two immortalized human LEC lines derived from LECs of different age.

## 2. Materials and Methods

### 2.1. Computational Methods for Estimation of Eye Temperature

Our computational techniques integrated the thermodynamics and the thermoregulatory response for increased temperature. When we considered the solar radiation, we also took into account its deposition in the human body by using electromagnetic computation. Detailed explanations and validations of our computational code are provided in our previous reports [13,14]. The effects of aging (>65 years old) on thermoregulation are a deterioration of the skin’s thermal sensitivity and a reduction of sweating by the limbs [15,16]. We applied two computational models for thermoregulation, which were validated in [17], for younger adult and elderly (>65 years old) test subjects.

A three-dimensional Japanese male model created based on magnetic resonance images was used in our computation [18]. The height and weight of this model were 1.73 m and 65 kg, respectively, which are close to the mean values of Japanese males. This model was segmented into 51 tissues including five tissues around the eye: lens, vitreous, iris, cornea, and anterior chamber. Figure 1 shows the three-dimensional human model and its enlarged figures around the eye.

### 2.2. Primary Culture of Human Lens Epithelial Cells

At the time of cataract surgery in patients who agreed to participate in this study, LECs that had adhered to the collected anterior capsule pieces were subjected to primary culture by the explant method. The composition of the culture medium was as follows: 10% fetal bovine serum (FBS), 10 ng/mL basic fibroblast growth factor (bFGF), ×100 minimum essential medium non-essential amino acids (MEM-NEAA), ×100 GlutaMAX™ (Thermo Fisher Scientific, Waltham, MA, USA), Dulbecco’s modified Eagle’s medium (DMEM) high-glucose medium, and penicillin-streptomycin solution (Sigma-Aldrich, St. Louis, MO, USA). This study was approved by the Ethics Review Committee of Fujita Health University (No. 004) and was conducted in accord with the provisions of the Declaration of Helsinki for research involving human tissue. As a control for immortalized human LECs, SRA01/04 strain cells (RCB1591, Riken BRC, Tsukuba, Japan) were used and cultured in the medium as instructed by the cell bank.

### 2.3. Immortalization Gene Transfer to Pseudo-Attachment (att) P Site

As the immortalizing gene, we used the modified SV40 Large T antigen-GFP (mSV40-GFP) synthesized for the production of the immortalized human corneal epithelial cell line as reported [19]. We used the Jump-In™ Gateway™ Expression System (Thermo Fisher Scientific) to prepare the donor plasmid by phiC31 integrase. Briefly, the cytomegalovirus (CMV) promoter of mSV40-GFP, poly-A SV40, and an empty entry plasmid vector were treated with restriction enzymes and ligated. The CMV promoter of the entry plasmid vector with ligated *attL1* and *attL2* was replaced with the Elongation Factor-1α (EF1α) promoter (Takara Bio, Shiga, Japan). The pJTI^TM^ Fast DEST plasmid vector carrying phiC31 with *attR1*, *attR2*, and the hygromycin resistance gene was then subjected to the Gateway LR reaction using LR clonase, and the Jump-In expression clone plasmid was introduced in *Escherichia coli* DH5α competent cell (Takara Bio). The growth and expression clone plasmids were then purified and transfected into primary human LECs by electroporation. The selection was performed with 200 μg/mL hygromycin B, and drug-resistant cells were cloned to establish an immortalized human lens epithelial cell line clone NY2 (iHLEC-NY2). The composition of the iHLEC-NY2 medium was as follows: 10% FBS, 2 ng/mL bFGF, ×50 GlutaMAX^TM^ I, Advanced DMEM/F12 medium (Thermo Fisher Scientific), and penicillin-streptomycin solution. This gene-recombination experiment was approved by the Kanazawa Medical University Biosafety Committee for Recombinant DNA Research (Approval No. 2020-18).

### 2.4. Temperature Maintenance during Medium Replacement and Cell Growth Curve

#### 2.4.1. Temperature Maintenance during Medium Replacement

While replacing the medium for the cells cultured at 37.5 °C and 35.0 °C, pre-warmed medium was used so as not to subject the cells to temperature changes. A culture dish was placed on a thin-type thermostatic plate (LABOPAD H; TOSC Japan, Tokyo, Japan) inside a clean bench to maintain the temperature of the medium. Then, a digital thermometer (CT-1310D; CUSTOM Corporation, Tokyo, Japan) connected with a sensor (TS-01WP; CUSTOM) with a diameter of 2 mm (Figure 2) was used to record the temperature of the medium every 15 s for 5 min.

#### 2.4.2. Cell Growth Curve

The cells were seeded at a density of 2.5 × 10^4^ cells per cell culture dish (60-mm dia., Eppendorf, Hamburg, Germany). The cells were cultured for 7 days at 37.5 °C, 35.0 °C, or 31.0 °C in 5% CO_2_ in a humidified incubator, and the medium was replaced every other day. The cells were counted in a Bürker–Türk counting chamber at 1:1 dilution with trypan blue solution (Sigma-Aldrich).

### 2.5. Detection of Mycoplasma Infection by Polymerase Chain Reaction (PCR)

As iHLEC-NY2 are primary cultured cells of human LECs, the detection of Mycoplasma infection was accomplished with a mycoplasma detection kit (EZ-PCR^TM^ Mycoplasma Test Kit, Biological Industries, Cromwell, CT, USA) according to the manufacturer’s instructions.

### 2.6. Quantitative Reverse Transcription-PCR

Total RNA of cells was extracted using an RNA extraction kit (RNeasy^®^ Micro; Qiagen, Tokyo). The RNA concentrations were measured using a spectrophotometer (NanoVue^TM^; GE Healthcare, Buckinghamshire, UK) [20].

Total RNA was reverse-transcribed using the GeneAmp^®^ polymerase chain reaction (PCR) system 9700 Thermal Cycler (Thermo Fisher Scientific) to synthesize complementary DNA (cDNA) [21]. Subsequently, quantitative real-time PCR (qPCR) was performed using the ABI PRISM^®^ 7900 HT Sequence Detection System (Thermo Fisher Scientific) with the Primer and Probe for the crystalline lens-marker genes *αA crystalline* (Hs00166138_m1, CRYAA) and *βB2 crystalline* (Hs00166761_m1, CRYBB2), as well as *glyceraldehyde-3-phosphate dehydrogenase* (GAPDH, Hs99999905_m1) as an internal positive control.

The qPCR of the Aβ marker genes amyloid precursor protein (APP), a disintegrin and metalloproteinase domain 10 (ADAM10), β-site APP-cleaving enzyme 1 (BACE1), presenilin 1 (PS1), presenilin 2 (PS2), neprilysin (NEP), and endothelin converting enzyme 1 (ECE-1) was performed using SYBR^®^ Green I (Roche Molecular Systems, Indianapolis, IN, USA) and specific primers as listed in reference [12].

We used the differences in the threshold cycles for GAPDH and target genes (CRYAA, CRYBB2, APP, ADAM10, BACE1, PS1, PS2, NEP, and ECE-1) to calculate the levels of mRNA.

### 2.7. Western Blotting

Cell lysates were prepared in ice-cold radioimmune precipitation (RIPA) buffer with the protease inhibitor phenylmethylsulfonyl fluoride (PMSF, FUJIFILM Wako Pure Chemical Corp., Osaka, Japan) and cOmplete™ Mini (Sigma-Aldrich) as described [22]. Equal amounts of protein samples were electrophoresed by sodium dodecyl sulfate-polyacrylamide gel electrophoresis (SDS-PAGE) and transferred to a polyvinylidene difluoride (PVDF) membrane (Millipore, Billerica, MA, USA). The membrane was incubated for 1 h at room temperature in a blocking buffer consisting of tris-buffered saline (TBS, 20 mM Tris-HCl pH 7.4, 137 mM NaCl) containing 3% skim milk. The membrane was then incubated overnight at 4 °C with the designated primary antibody: anti-human αA-crystalline mouse monoclonal antibody (1:1000, Santa Cruz Biotechnology, Santa Cruz, CA, USA) or anti-human βB2-crystalline rabbit polyclonal antibody (1:1000, Proteintech, Rosemont, IL, USA) or anti-human β-actin mouse monoclonal antibody (1:5000, Abcam, Cambridge, UK). The membrane was further incubated with horseradish peroxidase-linked secondary antibodies (Agilent Technologies, Santa Clara, CA, USA). Immunoreactive proteins were visualized with the Enhanced Chemiluminescence (ECL) detection system (GE Healthcare Bio-Science Japan, Tokyo, Japan). Quantification was performed using ImageJ software (U.S. National Institutes of Health, Bethesda, MD, USA).

### 2.8. Measurement of Human Aβ1-40 and Aβ1-42 by Enzyme-Linked Immunosorbent Assay (ELISA)

The culture medium obtained by culturing iHLEC-NY2 and SRA01/04 cells for 2 days was collected and then centrifuged at 1500× *g* for 5 min, and the supernatant was stored at −80 °C until measurement. The concentrations of Aβ1-40 and Aβ1-42 were measured using a Quantikine^®^ Enzyme-Linked Immunosorbent Assay (ELISA) kit (Research and Diagnostic Systems, Minneapolis, MN, USA) according to the manufacturer’s manual.

### 2.9. Statistical Analyses

Each experiment was performed in triplicate and repeated at least three times. Data are presented as the mean ± standard deviation (SD) and were analyzed by Welch’s *t*-test. The Statistical Package for Social Science (SPSS) Statistics 24 (IBM, New York, NY, USA) was used to perform the statistical analyses.

## 3. Results

### 3.1. Computational Results for the Estimation of Eye Temperature

The eye temperatures in younger and elderly adults with and without solar radiation were evaluated at ambient temperatures of 19–35 °C. The relative humidity was set at 50% considering the typical ambient condition. The effect of relative humidity is not high in a resting condition [23,24]. For test subjects exposed to solar radiation, an additional heat load obtained from the electromagnetic computation was applied. The incident power density of solar radiation integrated over the whole spectrum is the range of 250–750 W/m^2^ [25]. Considering the ambiguity of the incident angle to the eye, the value was averaged over the azimuth direction and determined as 150 W/m^2^.

The temperature distribution in the whole-body model is given at the steady-state in each scenario. The time required to reach a steady-state was 30 min (90% saturation). Figure 3 illustrates the eye temperature distribution in the depth direction with the corneal surface as x = 0. The temperature range of the lens over daily environmental changes is 35–37.5 °C. As shown in Figure 3, the difference in eye temperature observed at ambient temperatures above 30 °C, due to aging or the degradation of sweating.

### 3.2. Temperature Change in the Medium

Measurement of the medium temperature was performed under the following conditions: laboratory temperature, 25.3 °C; the temperature inside the clean bench, 29.3 °C, and surface temperature of the working table, 25.0 °C.

Figure 4 shows the temperature changes in the medium. The surface temperatures of the individual heat plates were 31.5 °C, 32.0 °C, and 34.7 °C, while the temperature of the heat plate device was set at 35.0 °C, 37.5 °C, and 40.0 °C, respectively. Under these conditions, the medium temperatures at 5 min after placing each dish cultured at 35.0 °C, 37.5 °C, and 40.0 °C were 32.7 °C, 34.1 °C, and 35.2 °C, resulting in a decrease of 1.9 °C, 3.0 °C, and 1.7 °C, respectively, compared to the baseline. When the dishes cultured at 35.0 °C and 37.5 °C were placed on the working table inside the clean bench, the medium temperatures at 5 min were 25.6 °C and 28.7 °C, which were 8.7 °C and 8.0 °C lower than the baseline, respectively.

On the other hand, the surface temperature of the heat plate was 36.9 °C when the device temperature was set at 42.5 °C, and the medium temperature at 5 min was 37.0 °C, representing a decrease of only 0.1 °C, which showed that the plate stayed warm even inside the clean bench.

### 3.3. The Cell Morphology and Proliferation of the iHLEC-NY2 Cells

The morphology of the iHLEC-NY2 cells was a short spindle shape (Figure 5a,b). The iHLEC-NY2 cells proliferated more actively compared to the SRA01/04 cells, and the proliferation of iHLEC-NY2 cells was significantly greater at the culture temperature 37.5 °C compared to 35.0 °C (Figure 5c, *p* < 0.005). The proliferation of SRA01/04 cells was also significantly greater at 37.5 °C (Figure 5c, *p* < 0.05). Moreover, the iHLEC-NY2 cells proliferated more actively compared to the SRA01/04 cells at the culture temperature of 37.5 °C (Figure 5c, *p* < 0.05). The iHLEC-NY2 cells hardly proliferated when the culture temperature was lowered to 31.0 °C (Figure 5d). The Mycoplasma infection result of the iHLEC-NY2 cells was negative (Figure 5e).

### 3.4. The mRNA Levels of iHLEC-NY2 and SRA01/04

In the iHLEC-NY2 cells, the gene expression of CRYAA was higher at 37.5 °C, but not significantly so. The gene expressions of APP, BACE1, NEP, and ECE-1 were significantly higher at 37.5 °C (Figure 6). In the SRA01/04 cells, the gene expressions of CRYAA, and ECE-1 were significantly higher at 37.5 °C (Figure 7).

In the comparison of the gene expressions in iHLEC-NY2 and SRA01/04 at 37.5 °C, CRYBB2 and ECE-1 were expressed more highly in iHLEC-NY2, while CRYAA, PS1, and PS2 were expressed more highly in SRA01/04. At 35.0 °C, CRYBB2 was expressed more highly in iHLEC-NY2, and PS2 and NEP were expressed more highly in SRA01/04 (Table 1).

### 3.5. Analysis of Western Blotting at CRYAA and CRYBB2

In both iHLEC-NY2 and SRA01/04 cells, the CRYAA bands were detected more strongly when the cells were cultured at 37.5 °C than at 35.0 °C. The integrated density (IntDen) of each captured band was analyzed and corrected with β-actin by the ImageJ analysis software. Most notably, CRYAA (which is a member of the small heat-shock protein family) was increased by ~1.37-fold in iHLEC-NY2 cells and by ~1.25-fold in SRA01/04 cells cultured at 37.5 °C compared to 35.0 °C (Figure 8).

### 3.6. Protein Concentrations of Human Aβ1-40 and Aβ1-42 in Culture Medium

Aβ1-40 and Aβ1-42 measured by ELISA are the protein concentrations contained in the medium (proteins released from cells, not cell lysates). Their values were therefore corrected by the number of cells when the medium was collected. The cell number-corrected concentration of Aβ1-40 was significantly increased when the cells were cultured at 37.5 °C compared to 35.0 °C (*p* < 0.05, Figure 9a). The cell number-corrected concentration of Aβ1-42 was not significantly different between 35.0 °C and 37.5 °C (Figure 9b). In SRA01/04 medium, both proteins were below the detection sensitivity of the ELISA kit.

## 4. Discussion

With computer simulations performed at ambient temperatures of 19 °C to 35 °C and a relative humidity of 50%, the estimated lens temperature was 35 °C to 37.5 °C. Therefore, when culture experiments were conducted using newly created human immortalized LECs (iHLEC-NY2) at culture temperatures of 35.0 °C and 37.5 °C, the gene expression at 37.5 °C was higher than that at 35.0 °C, and the amount of the synthesis of CRYAA (one of the major constituent proteins of the lens) increased. The Aβ-related genes and Aβ proteins secreted on the culture medium were also increased in iHLEC-NY2.

The lens temperature has been estimated as 35–37.5 °C depending on the ambient temperature of the eyeball; however, when the ambient temperature exceeds 30 °C, the estimated temperature of the lens differs depending on the body’s age, and it is higher with aging [13]. This may be due mainly to the difference in body temperature. In the elderly, the body temperature becomes higher and the perspiration amount decreases due to the decrease in the skin’s heat sensitivity. It can thus be inferred that the lens temperature may be more influenced by aging.

In this study, the culture temperature was the key factor. Because it was necessary to replace the medium regularly in the cell culture experiment, any temperature changes were carefully monitored using a sensor connected to a thermometer. Since the room temperature in the laboratory and that inside the clean bench were lower than the culture temperature, even the dish replacement from the incubator into the clean bench caused a slight decrease in the dish temperature. Moreover, the downflow of clean air in the clean bench also caused a decrease in the dish temperature. The equipment used (LABPAD H) has a two-stage structure, consisting of an adapter for the aluminum-block culture dish placed on a control unit with the sensor. The temperature difference between the set temperature of the device and the surface of the heat plate is related to the air downflow and the position of the sensor in the clean bench. In this study, when setting the temperature of the device at 42.5 °C, it was possible to maintain a temperature of around 37 °C even while working on the clean bench. To investigate the effect of temperature on the cells, it was important to set the environmental temperature during the experiment using the sensor connected to the thermometer.

In our epidemiological studies, the prevalence of nuclear cataracts was significantly increased in tropical and subtropical regions [8]. Studies using computer simulations indicated that the estimated temperature increased further from the center of the lens (lens nucleus) to the posterior subcapsule, which coincided with the opacity area of nuclear cataract [13]. As discussed by Miranda et al. [2], it was assumed that the higher ambient temperature causes an earlier onset of presbyopia, and the prevalence of nuclear cataracts increases with increasing sclerosis of the lens nucleus. It was also reported that individuals who engage in specific occupations such as glass blowing, foundry work, and blacksmithing may be more susceptible to posterior cataracts due to exposure to infrared radiation in addition to high fever over a period of years [26].

The factors that have been reported to affect the prevalence of nuclear cataracts include smoking [27], temperature [4], and diet [28]. A cohort study revealed that smoking was associated with an increased risk of age-related cataracts, especially nuclear cataracts [29]. In a study concerning ambient temperature, the lens temperature of monkeys placed outdoors in the sun at 49 °C rose to 41 °C within 10 min [30]. The lens temperature of rabbits was decreased by 7 °C in a rearing environment maintained at 4 °C [31]. In another experiment using rabbits, there were significant relationships between the ambient temperature in the sun and the temperatures of the lens and the posterior chamber aqueous humor [30]. It has been indicated that a significant increase or decrease in the environmental temperature and a long-term stay in a higher temperature environment affect the intraocular environment in and around the lens, inducing age-related cataracts [13]. As a dietary effect, high intakes of protein, vitamin A, niacin, thiamine, and riboflavin reduced the prevalence of nuclear cataracts [32], and vitamin C intake had a significant effect on the incidence of age-related nuclear cataracts [33]. Regarding racial differences, Caucasians had a significantly higher prevalence of nuclear cataracts than non-Caucasians [34,35,36].

The heat shock proteins (HSPs) are created by a typical reaction of a protein to heat (temperature). Among them, the low-molecular-weight HSPs (15–30 kDa) are proteins induced by heat stress, and typical HSPs are Hsp27 (HSPB1) and αB crystallin (HSPB5), the latter of which is a protein consisting of 175 residues of amino acids, with the molecular weight of approx. 20 kDa. The crystalline lens contains a large amount of CRYAA (HSPB4), which has approximately 55% amino acid homology and similar functions and structures, in addition to αB crystallin. The heat stability and chaperone function of the aggregation of the α-crystallin protein composed of both subunits is deeply involved in maintaining the transparency of the lens [37,38,39,40,41,42]. In addition, when the balance of calcium ions (Ca^2+^) at the molecular level is disrupted, scattered particles with a high molecular weight fraction are formed [43,44], resulting in scattering (opacity) of light in the crystalline lens. A differential scanning calorimetry study of α-crystallin revealed two endothermic transitions [45]. The first transition occurs at 35–51 °C with a peak at 45 °C. At 45 °C, the transition has been shown to be biologically relevant [46]. At this temperature, α-crystallin undergoes minor changes in its tertiary structure as its hydrophobic surface is exposed [47,48].

Intracellular homeostasis is disturbed by slight changes in environmental temperature, and the αA crystallin expression level may increase as one of the suppression mechanisms. The expressed αA crystallin is likely to be responsible for suppressing the intracellular dysfunction caused by temperature changes. For investigations of the temperature responsiveness at the molecular level, a lens-derived cultured cell model that is particularly susceptible to heat stress and has a defense mechanism would be optimal.

It was reported that Aβ was detected in the lens nucleus and deep lens fiber cells [48,49,50,51]. Aβ in the lens is produced by peptide degradation from APP. APP is decomposed into soluble sAPPα and C-terminal C83 by ADAM10 of the ADAM family with α-secretase activity [52,53,54], but Aβ is not produced by this decomposition. A part of the APP is decomposed into soluble sAPPβ and C-terminal C99 by BACE1 with β-secretase activity [55]. C99 is decomposed into the APP intracellular domain (AICD) and Aβ by a complex of PS1 and PS2 which has γ-secretase activity to produce Aβ [56]. It has been reported that most of the Aβ produced is Aβ1-40, and the remaining 10% is Aβ1-42, which is highly hydrophobic and easily forms amyloid fibrils [57,58,59,60]. NEP and ECE-1, well-known enzymes that decompose Aβ in vivo, are involved in the regulation of the Aβ concentration [61,62,63]. Herein we observed that most of the Aβ-related gene expression was elevated at 37.5 °C in both the iHLEC-NY2 and SRA01/04 cells. In the iHLEC-NY2 cells, higher levels of Aβ1-40 and Aβ1-42 secreted into the medium were detected at 37.5 °C, suggesting that the activity of Aβ-related factors may increase with temperature. In contrast, the levels of Aβ1-40 and Aβ1-42 in the SRA01/04 cells were below the detection sensitivity. In prior experiments using cell lysates of SRA01/04 cells, the accumulation of Aβ caused oxidative stimulation with oxidative stress such as nitric oxide, resulting in the positive feedback mechanism that Aβ expression and accumulation upregulate [11]. In iHLEC-NY2 cells, the formed Aβ may have been sufficiently decomposed, expressed increasingly by the positive feedback mechanism, and secreted into the medium. As the original cells used to create iHLEC-NY2 and SRS01/04 cells were from crystalline lenses of adults and infants, respectively, it is possible that the difference in intracellular metabolic activities derived from each of these origins affected the results. In the future, we plan to use cell lysates examined by a high-sensitivity ELISA method.

The SRA01/04 cells [10], an immortalized human LEC line, have been used in many studies and numerous papers have been described them. SRA01/04 cells, which have a morphology similar to that of HLECs, show the expression of lens-specific proteins such as CRYAA and CRYBB2. This cell line was transfected with a Rous sarcoma virus promoter and SV40 Large T antigen in a pGEM^®^-3Zf (+) vector using a calcium phosphate transfection system. Ibaraki et al. reported that SV40 Large T antigen was not transferred into the regions of CRYAA and CRYBB2 because the SRA01/04 line was created by the gene transfer method at its establishment [10], however, it is unknown to which area in the whole genome and how much SV40 Large T antigen was transferred.

It is quite difficult to culture adult human primary LECs, and even if the primary culture is successful, it is difficult to use it for basic research that requires reproducibility, as the cell expression is limited. The newly created cell line iHLEC-NY2 is an immortalized human LEC line that does not affect the existing genomic sequence due to the recombination of a modified SV40 Large T antigen that retains the original cell function [19], specifically into the *pseudo-attP* site, a non-functional site on the genomic sequence. In addition, modified SV40 Large T antigen is a non-viral vector and its recombination is single-copied. Compared to SRA01/04 cells produced from the LECs of infants with retinopathy of prematurity, the original cells that created iHLEC-NY2 are human adult LECs, in which Aβ observed in age-related cataracts was detected, suggesting that iHLEC-NY2 cells may be the immortalized human LECs that retain the metabolic environment of adult human LECs. We are continuing to analyze the genes and proteins of iHLEC-NY2.

iHLEC-NY2 is a human immortalized lens epithelial cell line created by gene transfer in a safe manner. As the immortalizing gene is introduced into the pseudo-attP site rather than randomly inserted into a gene originally present in the lens epithelial cells, the genetic characteristics of the human lens epithelial cell are preserved. This is an important feature compared to the more conventionally used SRA01/04 cells. In a comparison of iHLEC-NY2 and SRA01/04, iHLEC-NY2 cells had more active cell proliferation, and iHLEC-NY2 cells, but not SRA01/04 cells, secreted Aβ1-40 protein into the culture medium. In the medium of SRA01/04, Aβ1-40 protein was below the level of detection sensitivity. Although it could not be conclusively determined in the present study, our results suggested that the mechanism underlying the expression of Aβ1-40 protein was as follows: Aβ is peptide-degraded from APP, and PS1 and PS2 produce Aβ1-40. In iHLEC-NY2, the gene expressions of PS1 and PS2 were significantly lower than that of SRA01/04, the concentration of Aβ1-40 was low, and the expressions of all these proteins were detectable by a commercially available ELISA. Based on our findings, we speculate that iHLEC-NY2 cells are established from adult human lens epithelial cells, and may retain the mechanism of Aβ production in the original human lens epithelial cells. Because it has been reported that the Aβ1-40 protein is detected in the crystalline lens with age-related cataract [64] and Aβ1-40 protein can be detected in iHLEC-NY2 even by a commercially available ELISA, these cells will be convenient for use in future research.

Several factors may be involved in age-related nuclear cataracts: connexins and aquaporins (AQPs) related to metabolite transport in the lens, transient receptor potential cation channel subfamily V member 1 (TRPV1) and transient receptor potential cation channel subfamily V member 4 (TRPV4) proteins involved in lens ion transport [65], the post-translational modification of various crystallin proteins in the deep lens cortex and nucleus [66,67], and the oxidation of the lens nucleus due to antioxidants’ inability to reach the lens nucleus [68]. We plan to investigate various post-translational modifications, the effects of culture temperature on the proteins related to intercellular transport, two- and three-dimensional cultures, and lens organ cultures.

It has been speculated that in vivo, the lens temperature rises due to a high ambient temperature, and the HLECs, which originally express very slowly [1], express and spread quickly in the germinative zone of the lens, and cell metabolism is activated. From a long-term perspective, the aging of HLECs and the reduction of lens tissue stem cells are promoted; various proteins such as crystallin protein are denatured, the homeostasis of the lens cortex located around the lens nucleus is unbalanced, and an abnormal substance transport to the lens nucleus is caused, resulting in the induction of oxidation of the lens nucleus. We hypothesize that oxidation of the lens nucleus may cause the sclerosis of the lens nucleus, leading to refractive error causing presbyopia and light scattering of the lens nucleus, and opacity in the lens nucleus.

## 5. Conclusions

In computer simulations, when the ambient temperature was 19–35 °C, the estimated lens temperature was 35–37.5 °C. When the newly prepared iHLEC-NY2 cells were cultured at 37.5 °C and 35.0 °C, the cells’ proliferation and lens-specific protein gene and protein expressions increased, and higher cell activity was observed with higher temperature. The expressions of proteins and some genes of Aβ detected in human cataracts were also increased at 37.5 °C, suggesting that HLECs and lens aging may progress longitudinally with higher temperatures. Our present findings thus indicated that the ambient temperature was one of the contributing factors of nuclear cataracts observed in our epidemiological studies.

## Figures and Tables

**Figure 1 cells-09-02670-f001:**
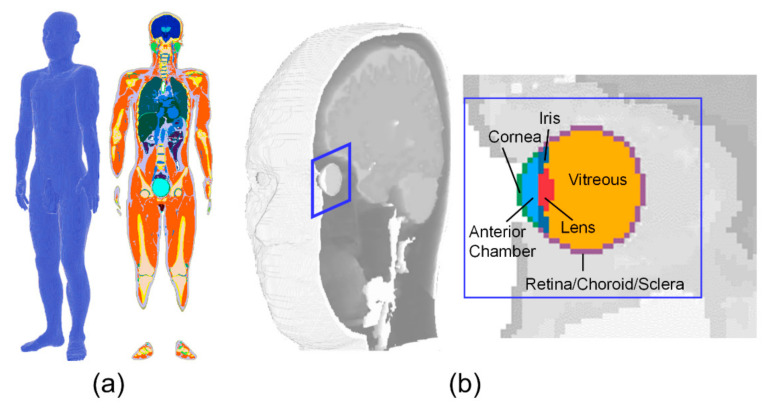
The human model and anatomy around the eye: (**a**) whole-body anatomical human model; (**b**) eye anatomy (derived from the whole-body model).

**Figure 2 cells-09-02670-f002:**
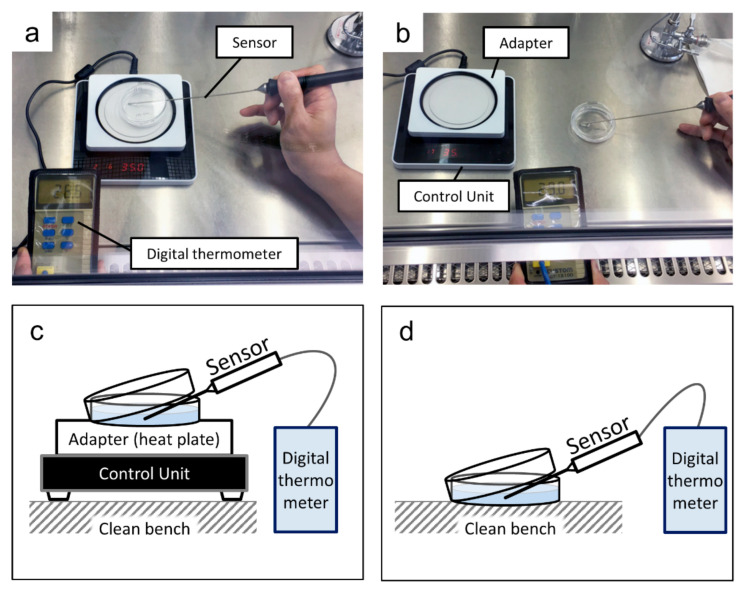
Measurement of the medium temperature: (**a**) the culture dish on a heat plate; (**b**) the culture dish on a working table inside a clean bench; (**c**) a schematic of the arrangement in (**a**); (**d**) a schematic of the arrangement in (**b**).

**Figure 3 cells-09-02670-f003:**
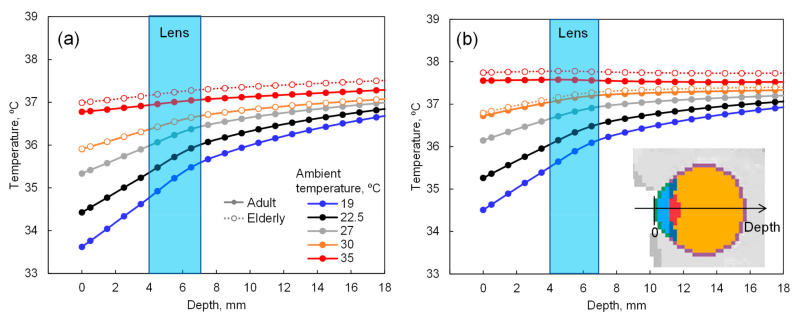
Eye temperature in the depth direction with the corneal surface as depth = 0 mm: (**a**) without and (**b**) with 150 W/m^2^ solar radiation.

**Figure 4 cells-09-02670-f004:**
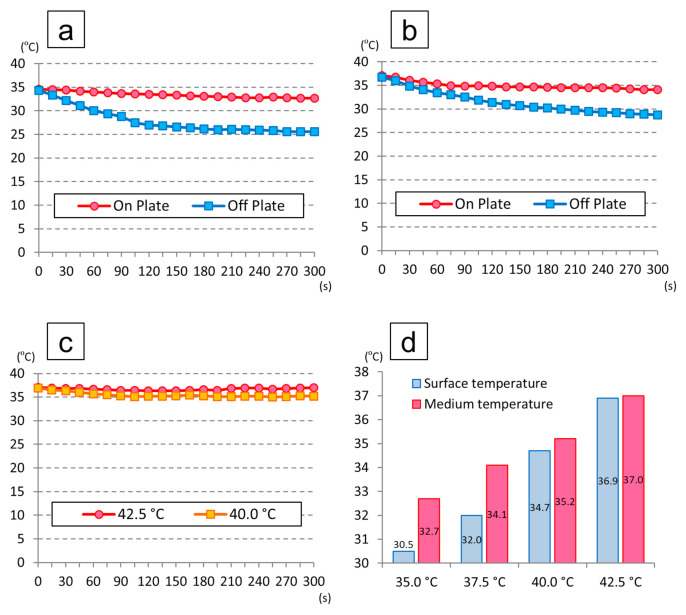
Temperature changes in the medium: (**a**) temperature changes of the dishes cultured at 35.0 °C in an incubator; (**b**) temperature changes of the dishes cultured at 37.5 °C in an incubator; (**c**) temperature changes when setting the temperature of the heat plate device at 40.0 °C and 42.5 °C; (**d**) temperatures of the adaptor surface and medium at 5 min when setting the temperature of the heat plate device at 35.0 °C, 37.5 °C, 40.0 °C, and 42.5 °C.

**Figure 5 cells-09-02670-f005:**
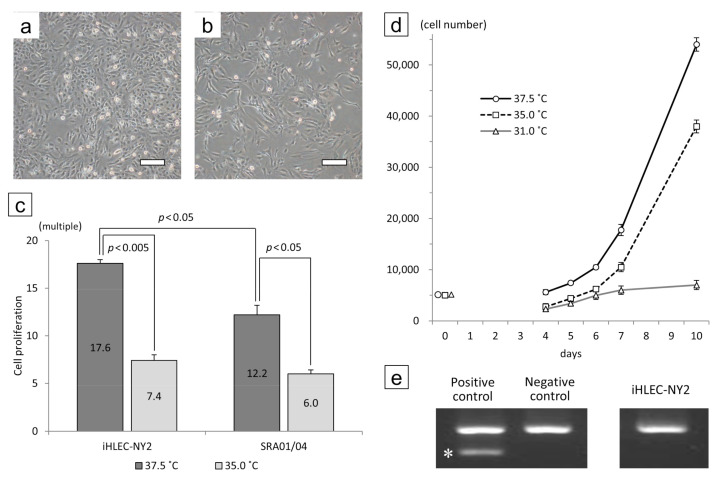
Cell morphology and growth rate of iHLEC-NY2 cells cultured at (**a**) 37.5 °C or (**b**) 35.0 °C for 7 days. Bar: 500 μm. (**c**) The ratio of cells grown after culturing for 7 days. (**d**) Growth curves of iHLEC-NY2 cells at different temperatures. (**e**) The result of the iHLEC-NY2 mycoplasma infection PCR test was negative. * Positive band.

**Figure 6 cells-09-02670-f006:**
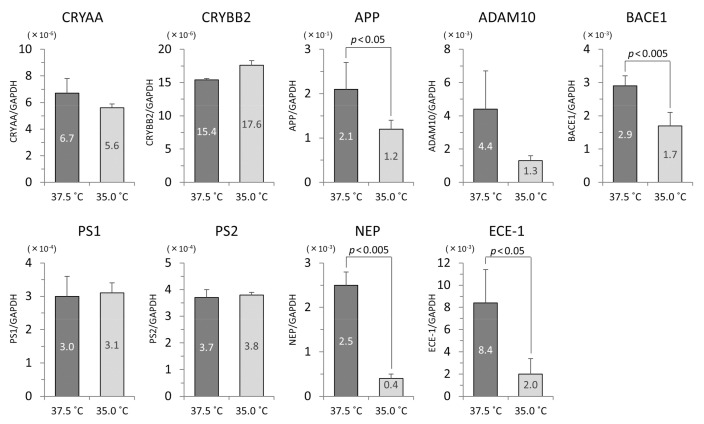
The qPCR results related to lens protein and Aβ protein in iHLEC-NY2 cells. The vertical axis is the mRNA value of the target gene-corrected by glyceraldehyde-3-phosphate dehydrogenase (GAPDH).

**Figure 7 cells-09-02670-f007:**
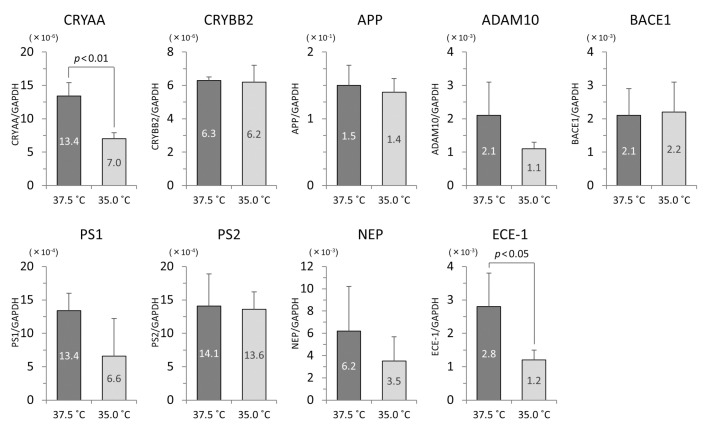
The qPCR results related to lens protein and Aβ protein in SRA01/04 cells. The vertical axis is the mRNA value of the target gene-corrected by GAPDH.

**Figure 8 cells-09-02670-f008:**
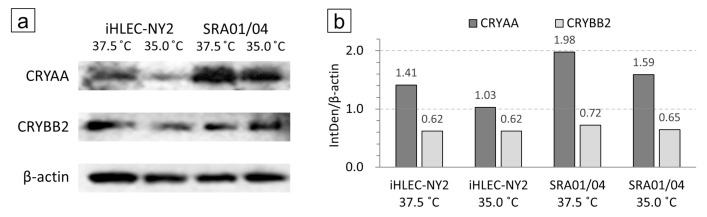
iHLEC-NY2 and SRA01/04 cell Western blotting results. (**a**) Immunostaining results of CRYAA and CRYBB2 in cells cultured at 37.5 °C and 35.0 °C. (**b**) The integrated density (IntDen) of each captured band was analyzed and corrected with β-actin.

**Figure 9 cells-09-02670-f009:**
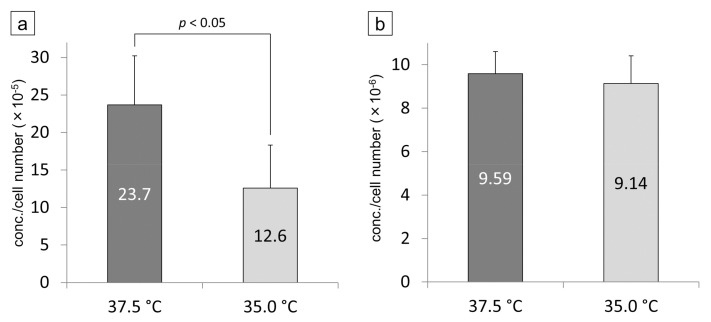
The proteins of Aβ1-40 (**a**) and 1-42 (**b**) in the iHLEC-NY2 culture medium were measured by ELISA. The detected concentration was corrected by the number of cells. The protein concentration of Aβ1-40 was significantly higher at 37.5 °C than at 35.0 °C (*p* < 0.05).

**Table 1 cells-09-02670-t001:** The comparison of the gene expressions in iHLEC-NY2 and SRA01/04 at 37.5 °C and 35.0 °C.

	37.5 °C			35.0 °C		
	iHLEC-NY2	SRA01/04	*p* Value	iHLEC-NY2	SRA01/04	*p*-Value
**CRYAA/GAPDH**	6.7 ± 1.1 (×10^−6^)	13.4 ± 2.0 (×10^−6^)	*p* < 0.05	5.6 ± 0.3 (×10^−6^)	7.0 ± 0.9 (×10^−6^)	n.s
**CRYBB2/GAPDH**	15.4 ± 0.1 (×10^−6^)	6.3 ± 0.1 (×10^−6^)	*p* < 0.001	17.6 ± 0.7 (×10^−6^)	6.2 ± 1.0 (×10^−6^)	*p* < 0.01
**APP/GAPDH**	2.1 ± 0.6 (×10^−1^)	1.5 ± 0.3 (×10^−1^)	n.s	1.2 ± 0.2 (×10^−1^)	1.4 ± 0.2 (×10^−1^)	n.s
**ADAM10/GAPDH**	4.4 ± 2.3 (×10^−3^)	2.1 ± 1.0 (×10^−3^)	n.s	1.3 ± 0.3 (×10^−3^)	1.1 ± 0.2 (×10^−3^)	n.s
**BACE1/GAPDH**	2.9 ± 0.3 (×10^−3^)	2.1 ± 0.8 (×10^−3^)	n.s	1.7 ± 0.4 (×10^−3^)	2.2 ± 0.9 (×10^−3^)	n.s
**PS1/GAPDH**	3.0 ± 0.6 (×10^−4^)	13.4 ± 2.6 (×10^−4^)	*p* < 0.05	3.1 ± 2.7 (×10^−4^)	6.6 ± 5.6 (×10^−4^)	n.s
**PS2/GAPDH**	3.7 ± 0.3 (×10^−4^)	14.1 ± 4.8 (×10^−4^)	*p* < 0.01	3.8 ± 0.9 (×10^−4^)	13.6 ± 2.6 (×10^−4^)	*p* < 0.001
**NEP/GAPDH**	2.5 ± 0.3 (×10^−3^)	6.2 ± 4.0 (×10^−3^)	n.s	0.4 ± 0.1 (×10^−3^)	3.5 ± 2.2 (×10^−3^)	*p* < 0.05
**ECE-1/GAPDH**	8.4 ± 3.0 (×10^−3^)	2.8 ± 1.0 (×10^−3^)	*p* < 0.05	2.0 ± 1.4 (×10^−3^)	1.2 ± 0.3 (×10^−3^)	n.s

n.s. = not significant.

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
