# Peer review of "Effect of a Lens Protein in Low-Temperature Culture of Novel Immortalized Human Lens Epithelial Cells (iHLEC-NY2)"

_cells, 2020, doi:10.3390/cells9122670_

Round 1

Reviewer 1 Report

Dear researchers,

                         I believe that this article is good, but according to my opinion needs a more extensive conclusion and in order to do this, i think that the best idea would be to made a table that compares your results with other results and after this table it would be need it to explain why your results are better from others.

In addition I feel that you need to explain all the abbreviations in your article, like iHLEC-NY2 cells for instance. This usually happened with two ways either you would explain in first use or (and i think is the best) you should made a table with abbreviations.

As for the end and very significant, it would be need it to make a figure that explains how your sensor works.

Kind regards

Reviewer 2 Report

Review of “Effect of lens protein in low-temperature culture under monitoring temperature maintenance of novel immortalized human lens epithelial cells (iHLEC5 NY2) using a temperature sensor”

This is an original research article. In the present study, the authors evaluated the relationship between the environmental temperature and the lens temperature by conducting a computer simulation in silico and they examined the effects of different temperatures on the lens, in vitro, using immortalized LECs at the lens temperature of residents of tropical and temperate regions.

Title: It needs to be modified in order to be more comprehensible.

Introduction: The background of the study is clear and the aim is well explained. Some sentences are too long and need to be modified. In paragraph 3, citations are missing regarding the “few reports showing the relationship between UV-B and nuclear cataracts”

Materials and Methods: The descriptions of computational methods for estimation of eye temperature, temperature maintenance during medium replacement, and culture of human lens epithelial cells are clear. A photo of the cell culture would be interesting. PCR was conducted according to the manufacturer's instructions. The concentrations of Aβ1-40 and Aβ1-42 were measured using a Quantikine® ELISA kit according to the manufacturer's manual. Each experiment was repeated and for statistical analysis Welch’s t-test on SPSS was used.

Results: The results are well presented and clear. Figures are accurate.

Discussion: Some citations should be added. (i.e. in the 7thparagraph, “It was reported…”). More information about a-crystallin and its key role in cataractogenesis is necessary. In the last paragraph, more data should be added regarding the possible explanation of the relation between cataract formation and temperature.

Conclusion: The authors clearly addressed their research question and stated that the cells' proliferation and lens-specific protein gene and protein expressions increased and higher cell activity was observed with higher temperature.

Overall, this research article presents an innovative topic in a comprehensive way. Citations are missing at some points, punctuation and syntax need improvement.

Round 2

Reviewer 1 Report

Dear researchers

I think that this manuscript is better and all the correction that i ask happened so from me is yes and it is accepted for publication

Kind regards

This manuscript is a resubmission of an earlier submission. The following is a list of the peer review reports and author responses from that submission.